# Lycopene Improves Metabolic Disorders and Liver Injury Induced by a Hight-Fat Diet in Obese Rats

**DOI:** 10.3390/molecules27227736

**Published:** 2022-11-10

**Authors:** Lina Baz, Salha Algarni, Mona Al-thepyani, Abdullah Aldairi, Hana Gashlan

**Affiliations:** 1Department of Biochemistry, Faculty of Science, King Abdulaziz University, Jeddah 21589, Saudi Arabia; 2Department of Chemistry, College of Science and Art, King Abdulaziz University, Rabigh 21911, Saudi Arabia; 3Department of Laboratory Medicine, Faculty of Applied Medical Sciences, Umm Al-Qura University, Makkah 24211, Saudi Arabia

**Keywords:** lycopene, high-fat diet, obesity, antioxidants, inflammatory, lipid peroxidation

## Abstract

Epidemiological studies have shown that the consumption of a high-fat diet (HFD) is positively related to the development of obesity. Lycopene (LYC) can potentially combat HFD-induced obesity and metabolic disorders in rats. This study aimed to investigate the effect of LYC on metabolic syndrome and assess its anti-inflammatory and antioxidant effects on the liver and adipose tissue in rats fed an HFD. Thirty-six male Wistar albino rats were divided into three groups. Group Ι (the control group) was fed a normal diet, group ΙΙ (HFD) received an HFD for 16 weeks, and group ΙΙΙ (HFD + LYC) received an HFD for 12 weeks and then LYC (25 mg/kg b.wt) was administered for four weeks. Lipid peroxidation, antioxidants, lipid profile, liver function biomarkers, and inflammatory markers were determined. The results showed that long-term consumption of an HFD significantly increased weight gain, liver weight, and cholesterol and triglyceride levels. Rats on an HFD displayed higher levels of lipid peroxidation and inflammatory markers. Moreover, liver and white adipose tissue histopathological investigations showed that LYC treatment mended the damaged tissue. Overall, LYC supplementation successfully reversed HFD-induced changes and shifts through its antioxidant and anti-inflammatory activity. Therefore, LYC displayed a therapeutic potential to manage obesity and its associated pathologies.

## 1. Introduction

Obesity is one of the most significant public health problems affecting populations worldwide. The concern is not about the extra fat tissue gained but the major health consequences accompanying obesity, such as dyslipidemia, cardiovascular diseases, type 2 diabetes, hypertension, and certain cancers, contributing to an increased risk of mortality as well as a reduced life expectancy and nonalcoholic fatty liver disease (NAFLD) [1,2,3,4,5,6]. During the past decades, experts, policymakers, educators, scientists, and health professionals have attempted to combat the obesity epidemic by designing efforts and strategies to raise awareness. Even so, unfortunately, it has been increasing dramatically worldwide [7,8]. The etiology of obesity is complex and involves many factors that interact with one another, such as dietary patterns, sedentary lifestyle, socioeconomic status, genetics, and psychological profile [3]. Epidemiological studies have demonstrated that eating a high-fat diet (HFD) is positively associated with the development of obesity. An HFD induces the overconsumption of calories leading to weight gain and fat accumulation [9].

### Obesity: Treatments, Conceptualizations, and Future Directions for a Growing Problem

NAFLD has become one of the leading causes of chronic liver diseases in the industrialized world, with an estimated global prevalence of 25–30%, rising to 90% in morbidly obese patients [10]. Monitoring of markers of liver function, such as aspartate aminotransferase (AST), alanine aminotransferase (ALT), alkaline phosphatase (ALP), and albumin, and markers of lipid metabolism such as triglycerides (TG), cholesterol, high-density lipoprotein (HDL), low-density lipoprotein (LDL), and very low-density lipoprotein (VLDL), as well as abdominal ultrasound, are the most commonly used methods for assessing various diseases including NAFLD risk [11,12,13,14]. However, since the early stages of NAFLD usually disclose no apparent symptoms, the prevalence of obesity-driven NAFLD and consequent morbidity can be considered one of the main health crises of the next decade [8,10,15].

Outside the liver, adipose tissue can secrete various bioactive peptides that produce multiple effects at local and systemic levels [16,17,18,19]. In the NAFLD state, enlarged adipocytes appear to secrete more adipokines. It has been observed that adipose tissue secretes about fifty adipokines, which impact inflammation and body weight homeostasis. These include leptin, adiponectin, tumor necrosis factor-alpha (TNF-α), interleukin 6 (IL-6), and resistin [17,18,20,21]. Some markers of oxidative stress and antioxidants have been studied to assess the redox state in NAFLD. Oxidative stress biomarkers including lipid peroxides, malondialdehyde (MDA) and nitric oxide (NO) had increased activities in most NAFLD clinical models [22]. Additionally, NAFLD decreased the activities of antioxidant markers, superoxide dismutase (SOD) and catalase (CAT) in patients with NAFLD [23,24]. Plant-based food is classically associated with fighting obesity due to its macronutrient composition and micronutrients, such as carotenoids [25,26,27]. Lycopene (LYC), a red-colored carotenoid, has attracted the interest of nutritionists, medical experts, and researchers because it can be used in the treatment of various human diseases such as cancer, diabetes, obesity, cardiovascular disease, and respiratory disease [28,29,30,31]. Before intestinal absorption, ingested LYC is emulsified and solubilized into micelles. They are contained in chylomicrons for transportation to the liver and secreted into the lymphatic system. Carotenoids are stored or secreted by the liver in LDL and VLDL in the fed state. Plasma carotenes are mainly present in LDL in a fasting state. LYC is converted into apo-10-lycopenoid and apo-lycopenbic acid by carotene 9,10-oxygenase, a mitochondrial enzyme found in the liver and other organs [32]. These metabolites are transported into peripheral tissues to perform their biological functions [33,34].

LYC had a protective role in a rat model of NAFLD by lowering liver enzyme levels, such as AST, ALT, ALP, and albumin, and lipid metabolites such as TG, cholesterol, HDL, LDL, and VLDL [34,35,36,37,38]. LYC could protect redox homeostasis and inhibit the overproduction of proinflammatory cytokines [39]. Several studies have reported that LYC has a modulating effect by reducing TNF-α, IL-6, leptin, adiponectin, resistin, and oxidative stress markers such as MDA and NO. Moreover, studies have also reported that it elevates the levels of the antioxidant enzymes SOD and CAT in the livers of obese mice [34,40,41]. The present study aimed to consider the potential effect of LYC on controlling obesity and its adverse sequelae by assessing liver function parameters, lipid profiles, inflammatory markers, oxidative stress biomarkers, and antioxidant enzymatic activities. Furthermore, a histopathological study was performed on the liver and white adipose tissue of rats fed an HFD.

## 2. Results

### 2.1. Effect of LYC on Body Weight and Abdominal Fat

The food intake and liver weight index of the HFD group and HFD + LYC rats recorded a significant decrease (*p* < 0.05) compared to the control rats. In addition, significant increases were observed in weight gain, abdominal fat, and the abdominal fat index (*p* < 0.05) in the HFD and HFD + LYC rats relative to the control rats. Furthermore, HFD + LYC rats presented a significant decline (*p* < 0.05) in the abdominal fat index. Additionally, no significant difference was observed in liver weight between groups (Table 1).

### 2.2. Effect of LYC on Liver Function Biomarkers

Rats fed an HFD displayed notable elevations (*p* < 0.05) in AST, ALT, and ALP levels compared to those who received a normal diet. Meanwhile, the HFD + LYC group showed no significant difference in the serum level of AST and a notable reduction in ALT and ALP levels (*p* < 0.05) compared to the control group. Additionally, HFD + LYC rats showed a significant decrease (*p* < 0.05) in AST, ALT levels and no significant change in ALP levels relative to rats in the HFD group. Further, the serum level of albumin was reduced significantly in rats fed HFD compared to the control group. There was no significant change in albumin in HFD + LYC rats compared to the control and HFD groups (Figure 1).

### 2.3. Effect of LYC on Lipid Profiles

Marked rises (*p* < 0.05) were noticed in serum levels of TG, cholesterol, and VLDL. However, there was no significant change in the level of HDL and LDL in rats submitted to an HFD compared to those fed a normal diet. TG and VLDL serum levels showed notable elevations in the HFD + LYC group compared to the control group. There was no significant difference in the serum level of cholesterol in the HFD + LYC when compared to rats on a normal diet. A significant increase (*p* < 0.05) in the serum level of HDL was observed in the HFD + LYC group compared to the control and HFD groups. In contrast, no statistically significant difference was noted in the HFD group compared to the control group. Additionally, a significant decrease (*p* < 0.05) was observed in the serum level of LDL in HFD + LYC rats compared to rats fed an HFD, although there was no significant difference in HFD rats compared to the control groups (Figure 2).

### 2.4. Effect of LYC on Inflammatory Biomarkers

Levels of hepatic TNF-α, IL-6, leptin, and resistin exhibited notable increases (*p* < 0.05) in the HFD-administered group, while adiponectin was noted to show no significant difference compared to the control group. Additionally, no significant change was observed in the TNF-α, IL-6, and leptin of HFD + LYC rats compared to the group fed a normal diet. However, a remarkable decrease (*p* < 0.05) was observed in the adiponectin level. A significant increase in the level of resistin was found in the HFD + LYC group compared to the group fed a normal diet. A significant decrease was observed in TNF-α, IL-6, and leptin levels in the HFD + LYC group. There was no significant change in the hepatic adiponectin and resistin of the HFD + LYC group compared to HFD rats (Figure 3).

### 2.5. Effect of LYC on Hepatic Oxidative and Antioxidant Status

As shown in Figure 4, there was a significant increase (*p* < 0.05) in the level of MDA in HFD rats. There was no significant change in NO, SOD, and CAT levels compared to the normal control group. Likewise, there was no statistically significant change in the MDA, NO, SOD, and CAT levels in the HFD + LYC group compared to the normal group. Further, a notable decrease (*p* < 0.05) was observed in the level of MDA in the liver of the HFD + LYC group, while no statistically significant difference was shown in the level of NO compared with the HFD group. However, HFD + LYC rats exhibited marked antioxidant effects as witnessed by increases (*p* < 0.05) in the levels of SOD and CAT compared with the HFD group.

### 2.6. Histopathological Findings

#### 2.6.1. Liver

The livers of the control group were deep red, moist, shiny, and robust when seen with the naked eye (Figure 5a). In contrast, yellow necrotic foci, lackluster appearance, and swelling were observed in the HFD group (Figure 5b). Additionally, in HFD + LYC rats the liver was a bright red color without yellow necrotic foci (Figure 5c). In H&E-stained sections, in the HFD and HFD + LYC groups, the periportal area exhibited extensive steatosis. Hepatocytes contained macrovascular fat vacuoles (MAFVs) and microvascular fat vacuoles (MIFVs). MAFVs in certain hepatocytes pushed the nucleus to the periphery. Additionally, a ballooned hepatocyte with a Mallory body was present. Mononuclear inflammatory cells (ICs) and eosinophils infiltrated the entire zone. Congestion occurred in the central veins (CVs) and hepatic arteries (Figure 5e,f).

#### 2.6.2. White Adipose Tissue

The white adipose tissue (WAT) in the HFD group had larger fat cells due to the accumulation of stored fat in the form of lipid droplets that coalesced into a single large droplet that expanded and covered most of the cytoplasm (Figure 5h). In contrast, the control group had a normal adipocyte distribution and cell size (Figure 5g). Adipocyte sizes and histology were also reduced in the LYC cotreated rats, similar to the control group (Figure 5i).

## 3. Discussion

Obesity is considered a global public health concern; an effort is needed to understand both the reason behind the rapid increase in its prevalence and its relationship with chronic diseases. An HFD is often overconsumed which leads to an increased body weight in mammals [42]. The overproduction of free radicals and oxidative damage have been associated with developing dyslipidemia and related events; therefore, several studies have suggested that using natural antioxidants may be beneficial in HFD-related problems. In this sense, the present study investigated dietary supplementation with LYC in obese rats. We evaluated hepatic function by measuring liver enzymes and monitored the lipid profiles of the rats. The responses of inflammatory markers, oxidative stress, and the antioxidant defense system were also measured.

Obesity in humans is often caused by an imbalance between energy intake and consumption. It is defined by abnormal accumulation of body fat and persistent low-grade inflammation. In the current study, the feed intake of the HFD group was significantly lower than that of the control group. However, the increased fat content in the diet contributed to an increase in obesity and relative fat mass. The higher caloric intake in the HFD group may have promoted weight gain due to the increased fat mass [41]. These findings are supported by previous studies [43,44,45].

Consistent with previous findings, rats fed the HFD had significant elevations in serum AST, ALT, and ALP compared to those fed the standard diet [46,47]. Commonly, these enzymes are found in large quantities in the cytoplasm or mitochondria of liver cells, while in hepatopathy, they leak to the bloodstream. The degradation of hepatocyte membranes and the resulting loss of integrity and permeability may be responsible for the increased levels of serum enzymes in the HFD group [48,49]. These findings were confirmed by histopathological examination of the liver, which showed that the HFD group had centrilobular liver necrosis, balloon-like degeneration, dilated sinusoids with many fat bodies, hepatocyte vacuolation, and infiltrating lymphocytes [34,50]. In contrast, serum levels of AST and ALT in HFD rats coprotected with LYC were decreased. According to a study by Jiang et al. [50], concomitant treatment with LYC at a dose of 20 mg/kg significantly lowered blood levels of AST and ALT in a rat model of NAFLD. Additionally, mice fed a high-saturated-fat and high-cholesterol diet supplemented with dried tomato peel showed consistent results [51]. In addition, albumin levels were considerably lower in rats fed an HFD compared to the control group. Hepatocytes are primarily responsible for the production of albumin. Therefore, its decrease in serum is a marker for liver disease [41,52,53,54,55]. The decrease in serum albumin could be due to albuminuria, a condition associated with renal dysfunction caused by abdominal obesity [56,57]. Albumin is an antioxidant protein that decreases obesity-related inflammatory states. Consequently, the decrease in albumin levels promotes oxidative stress in obese rats [41,58,59].

The current study showed that rats fed an HFD had drastically altered lipid profiles, resulting in dyslipidemia in TG, cholesterol, and VLDL levels. This was also reported by [60,61,62]. Elevated TG in association with higher levels of LDL and lower levels of HDL shown in the HFD group is a characteristic of dyslipidemia in obesity. Another significant finding of our study was that LYC cotreatment resulted in considerable reductions in LDL levels in conjunction with increased HDL in obese rats. This finding is similar to that of previous research that found that LYC significantly enhances serum parameters of lipid metabolism [40,63,64]. Previous research has shown that LYC can affect cholesterol metabolism in several ways. The HDL metabolic pathway has revealed the main processes regulating HDL levels in the blood. Lecithin cholesterol acyltransferase (LCAT) and cholesteryl ester transfer protein (CETP) esterify HDL particles composed of lipids, proteins, and free cholesterol. In addition, these two proteins regulate the conversion of TG in VLDL and LDL to cholesterol ester in HDL [65]. McEneny et al. examined the effect of LYC on HDL-associated inflammation in moderately overweight, middle-aged individuals. They found that LYC supplementation reduced CETP and increased LCAT activities in the serum [66]. On the other hand, LYC decreases cholesterol synthesis through the reduction in 3-hydroxy-3-methyl-glutaryl-coenzyme A (HMG-CoA) reductase activity and modulation of the LDL receptor and acyl-coenzyme A transferase (ACAT) activity [67].

In addition, elevated levels of inflammatory cytokines TNF- α and IL -6 were found in the current study. Our results agree with those of [44,68,69]. According to previous studies, LYC has successfully produced anti-inflammatory effects and a hepatoprotective effect by lowering cytokine activity [40,70]. Apo-10’-lycopenoic acid, a metabolite of LYC, significantly reduces liver inflammation in mice fed an HFD by inhibiting the release of cytokines and lowering TNF, IL-6 and NF-B p65 [71]. Furthermore, increased fat mass in the present study promoted leptin and resistin levels, which is consistent with the findings of [72,73]. Leptin controls body weight and energy balance, curbs appetite and is often called the “satiety hormone” or “starvation hormone”. In an obesity state, leptin levels are higher, but leptin activity is lower due to leptin resistance. Increased leptin levels also positively affect the incidence of metabolic syndrome and cardiovascular disease [74]. In our study, the HFD effectively increased the leptin levels and weight gain, resulting in hyperleptinemia. Elevated levels indicate leptin resistance, which alters energy expenditure and causes lower food intake [75]. LYC coadministration significantly reduced hyperleptinemia by modulating epididymal fat mass and adipocyte circumference, and by reducing inflammatory cytokines that may down-regulate leptin in adipocytes [76].

Leptin has recently emerged as a critical link between metabolic responses and inflammation. It is thought that elevated levels of leptin in obese individuals can contribute to low-grade chronic inflammation, which could lead to degenerative diseases and autoimmune reactivity [77]. Our results confirm that body fat mass positively correlates with serum leptin levels and negatively correlates with adiponectin levels [78]. Adiponectin secretion inhibited by several factors, including a high level of TNF-α and oxidative stress [79]. Therefore, our data show decreased adiponectin levels in the HFD group, associated with increased levels of TNF-α and oxidative stress, confirming the correlation between adiponectin levels and an inflammatory state [80]. Accordingly, our data indicate that a high leptin/adiponectin ratio in HFD rats is associated with more elevated inflammation markers. The analysis of the additional proinflammatory adipokines showed that HFD raised the levels of resistin, and our results are corroborated by reports from other laboratories [74].

Resistin is a peptide hormone synthesized by mature adipocytes in mice, and by macrophages and monocytes in humans [81]. It induces and increases IL-6 and TNF-α in human monocytes and can directly suppress adiponectin’s anti-inflammatory activities [82]. Resistin secretion is elevated in obesity, and it plays a crucial role in activating inflammatory M1-type macrophages and synthesizing protective adiponectin in adipose tissue [74]. LYC coadministration reduced adipocyte hypertrophy in rats fed an HFD. Regarding the effects of LYC, this result is consistent with previous studies that found oral LYC supplementation reduced adipocyte hypertrophy in mice [83,84].

Increased metabolic changes in oxidation, the Krebs cycle and oxidative phosphorylation are used to fight lipids’ excessive formation in hepatic tissue. Deficient metabolic adaption, however, leads to excessive ROS production and oxidative stress during an extended period of overnutrition [40]. The current study showed a significant elevation in MDA in the HFD group [60,85,86,87]. Increased levels of MDA and severe lipid peroxidation, which are essential contributors to the development of oxidative stress and may harm the integrity of cellular membranes, were seen in the obese state [88,89]. Administering LYC effectively reduced the oxidative stress caused by obesity and increased antioxidant reserves. The antioxidant properties of LYC control the production of antioxidant enzymes such as SOD and CAT while directly scavenging free radicals [90,91,92,93]. In vitro studies show that LYC is two times more effective than β-carotene and ten times more effective than α-tocopherol in inhibiting singlet oxygen. In addition, it can interrupt the effects produced by free radicals, including radical peroxides and hydroxyl radicals [90,94,95]. LYC’s chemical composition and liposolubility may be associated with its ability to prevent diseases such as oxidative stress. It is an important protective factor integrated into cell membranes. In addition, it contains many double bonds, a characteristic that enhances its antioxidant capacity [96,97,98].

A study was conducted on hyperhomocysteinemic Wistar rats receiving lycopene (5 mg/kg) for three months, which showed a reduction in MDA concentrations indicating that lycopene could regulate redox imbalances [88], which is consistent with our findings. In an NAFLD model, SOD and CAT activities were elevated in Sprague Dawley rats after four-week lycopene (20 mg/kg) treatment, according to [86]. LYC decreases MDA and increases CAT activity in a rat model of hepatic ischemia and reperfusion [99]. Additionally, LYC administration in diethylnitrosamine-induced hepatocarcinogenesis in Wistar rats decreased MDA levels while increasing SOD and CAT activity [100]. In addition, Sheik Abdulazeez et al. and Imran et al. found that lycopene can reduce oxidative stress in a Wistar rat model of D-galactosamine/lipopolysaccharide-sensitized liver injury by de-creasing MDA levels and increasing antioxidant defenses such as CAT and SOD [37,101].

## 4. Materials and Methods

### 4.1. Animals and Ethical Statement

A total of thirty-six male albino rats (Wistar strain), weighing approximately 200 ± 10 g (~7 weeks old), were used in this study. The rats were randomly assigned into three groups, 12 rats per group (with four rats kept in each cage), and kept in a quiet, stress-free environment, with a temperature of 20–22 °C and humidity of 60%, on a 12 h light/dark cycle. Rats were supplied with standard pellet chow with free access to tap water for two days before the experiment for acclimatization. This study was approved by the King Abdulaziz University’s Ethics Committee (approval No. 518-20) and followed the Animal Care and Use Committee guidelines at King Fahad Medical Research Center (KFMRC).

### 4.2. Standard Animal Diet

A standard nutritionally balanced diet was obtained from KFMRC. The diet was composed of the following ingredients: crude protein 20.0%, crude fat 4.0 %, crude fiber 3.50%, mixed vitamins 1.0%, mixed minerals 3.50%, choline chloride 0.25%. The mixture was made up to 100% with corn starch, and its energy equaled 2850 kcal/kg. The diet was purchased from Grain Silos and Flour Mills Organization, KSA. Since protein and carbohydrate both contain 4 kcal/g, and fat contains 9 kcal/g, the 67.75 g of carbohydrate in our standard diet provided 271 kcal, the 20 g of protein provided 80 kcal, and the 4 g of fat provided 36 kcal. Hence, every 100 g of standard diet produced 387 kcal (271 + 80 + 36), where 70% of the calories are derived from carbohydrates, 20.6% from protein, and 9.3% from fat (percent by calorie).

### 4.3. Preparation of the High-Fat Diet (HFD)

DL-methionine, vitamins and mineral mix were added to overcome the limitation of casein and prevent vitamin and mineral dilution. For this purpose, a powdered normal-pellet diet (NPD) (1000 g), butter (500 g), casein (125 g), DL-methionine (3 g), and vitamin and mineral premixes (50 g) were thoroughly mixed to produce 1678 g of HFD. Since each 1000 g of NPD contained 677.5 g of carbohydrate, 200 g of protein, and 40 g of fat, by adding the above compounds to this base, the final amounts of fat, protein, and carbohydrate reached 540 g, 325 g, and 677.5 g, respectively. To acquire the weight percentage, if 1678 g of our HFD contains 540 g of fat, 325 g of protein, and 677.5 g of carbohydrate, then 100 g of this HFD would hence have 32.181 g of fat, 19.368 g of protein, and 40.375 g of carbohydrate [102].

### 4.4. Preparation of LYC

LYC (purity 98%, Xian Tongze Biotech Co., Ltd., Shanxi, China) was mixed with sunflower oil and stored at 4 °C in the dark until use. The tomato oleoresin–sunflower oil mixture was stirred for 20 min in a water bath at 37 °C before being fed to the animals. The dose of LYC was selected according to [103,104,105].

### 4.5. Experimental Design

The rats were divided into three groups, each with 12 rats:

Group Ι received a normal diet during the whole experimental period (12 weeks), and then, the rats were gavage-fed sunflower oil (~2 mL/kg /B. W/day) for the remaining four weeks.

Group ΙΙ received the HFD during the whole experimental period (12 weeks), and then the rats were gavage-fed sunflower oil (~2 mL/kg/B.W/day) for the remaining four weeks.

Group ΙΙΙ received the HFD throughout the experimental period (12 weeks) and then the rats were gavage-fed LYC (25 mg/kg/B.W/day) for the remaining four weeks. The weights of all rats from different groups were recorded during the experiment.

### 4.6. Sampling and Tissue Preparation

At the end of the experimental period, all rats fasted for 10 h, water was not constrained, and then blood samples were drawn under the effect of isoflurane inhalation anesthesia from the retroorbital venous plexus of the eyes into clean gel tubes. Blood samples were centrifuged at 3000 rpm for 15 min to separate serum and then divided into several aliquots and stored at −20 °C for biochemical analysis (Kilany et al., 2020). After that, the rats were sacrificed by an overdose of isoflurane. Each rat’s liver and adipose tissue were immediately enucleated, washed with buffered saline (0.9% NaCl solution), blotted with filter paper, and then weighed. A part of the liver and adipose tissue was immersed in a 10% paraformaldehyde solution for histopathological investigations. The remaining liver parts were kept at −80 °C for liver tissue homogenate preparation to estimate oxidative stress and antioxidant enzymes [106].

### 4.7. Serum Biochemical Parameters

#### 4.7.1. Liver Function Tests and Lipid Profile

The serum levels of aspartate transaminase (AST), alanine transaminase (ALT), alkaline phosphatase (ALP), albumin, triglycerides (TG), cholesterol, and high-density lipoprotein (HDL) were determined using standard kit methods using a fully automated COBAS^®^ 8000 modular analyzer series in the King Fahad Armed Forces Hospital, Jeddah, Saudi Arabia. The levels of LDL and VLDL were assessed according to Friedewald’s equation [107]:LDL = TC-HDL − (TG/5)
VLDL = TG/5

#### 4.7.2. Determination of Inflammatory Markers

The serum levels of tumor necrosis factor-α (TNF-α), interleukin-6 (IL-6), leptin, adiponectin, and resistin were assayed using Cat. No: SEKR-0009, Cat. No: SEKR-0005, Cat. No: SEKR-0051, Cat. No: SEKR-0063, and Cat. No: 0092 (Solarbio, Maju Bridge Town, Beijing, China), respectively.

### 4.8. Oxidative Stress Markers

To assess biomarkers for oxidative stress, malondialdehyde (MDA) and nitric oxide (NO) levels were measured using Cat No: BC0025 and Cat. No: BC1475 (Solarbio, Maju Bridge Town, Beijing, China), respectively.

### 4.9. Antioxidant Enzymatic Activities

Superoxide dismutase (SOD) and catalase (CAT) activities were determined using Cat No: BC0175 and Cat. No: BC0205 (Solarbio, Maju Bridge Town, Beijing, China), respectively. The procedures of 2.6.2, 2.7, and 2,8 were determined using ELISA kits followed according to the manufacturers’ enclosed pamphlet.

### 4.10. Histological Examination

Histological examinations were performed to evaluate the liver and fat deposition within the white adipose tissue. For this purpose, liver and adipose tissues were fixed in a 10% paraformaldehyde solution and then embedded in paraffin. The embedded tissue samples were sectioned (5 µm) and stained with hematoxylin and eosin (H&E) to examine general histological features.

### 4.11. Statistical Analysis

GraphPad Prism 9 was used to analyze all data, presented as the mean ± standard error of the mean. All data were compared between the groups using a one-way analysis of variance (ANOVA) and Tukey’s post hoc test. A value of *p* < 0.05 was used to determine statistically significant differences between groups.

## 5. Conclusions

In this study, we found that HFD consumption was associated with weight gain, fat accumulation in the abdomen and liver, elevated liver enzymes, dyslipidemia, and disturbances in the assessed inflammatory and oxidative stress parameters. The administration of LYC ameliorated the development of these abnormalities by restoring the levels of TG, cholesterol, HDL, LDL, and VLDL. In addition, treatment with LYC improved inflammation by lowering inflammatory biomarkers TNF-α, IL -6, and leptin. The reduction in MDA levels indicated an improvement in the lipid peroxidation status. Nevertheless, supplementation of LYC increased the activities of antioxidant enzymes, including CAT and SOD, in the livers of obese rats. Additionally, LYC amended obesity-induced hepatic injury. Histopathological examination of the adipose tissue and liver showed the ameliorative effect of lycopene, as evidenced by a reduction in adipocyte size and lower fat accumulation in the liver. Overall, the present study suggests that treatment of obese rats with LYC protected the body from the harmful effects of an HFD through its antioxidant and anti-inflammatory properties. Further studies on the effect of LYC on other HFD complications such as diabetes, cardiovascular disease, and kidney failure are recommended. Additionally, since obesity is caused by a defect in fat metabolism, it would be interesting to conduct more studies on the organelles responsible for lipid metabolism such as mitochondria and peroxisomes.

## Figures and Tables

**Figure 1 molecules-27-07736-f001:**
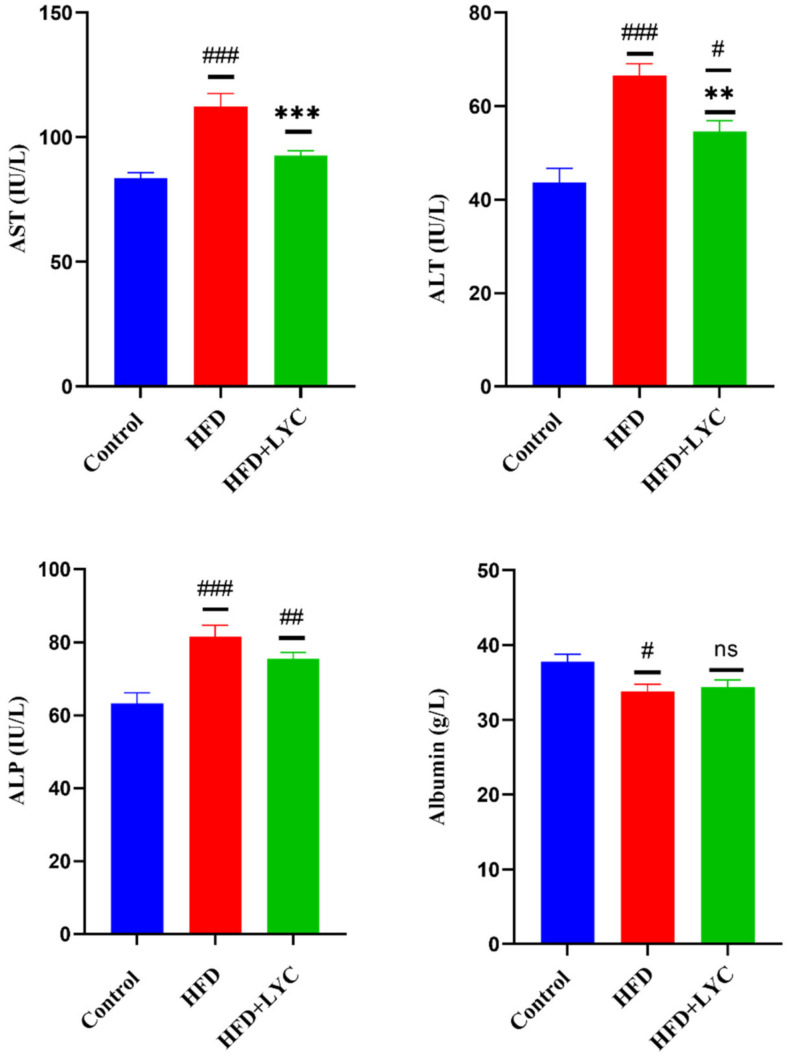
Serum levels of liver function biomarkers AST, ALT, ALP, and albumin following treatment with LYC in rats fed an HFD. Data are expressed as the mean ± SEM of two replica (*n* = 12 rats). Different markers correspond to statistically significant differences between groups. # *p* < 0.05, ## *p* < 0.01, and ### *p* < 0.001 indicate significant, highly significant, and very highly significant compared with the control, respectively. ** *p* < 0.05 and *** *p* < 0.001 indicate a highly significant and a very highly significant difference compared to the HFD group, respectively. ns: nonsignificant.

**Figure 2 molecules-27-07736-f002:**
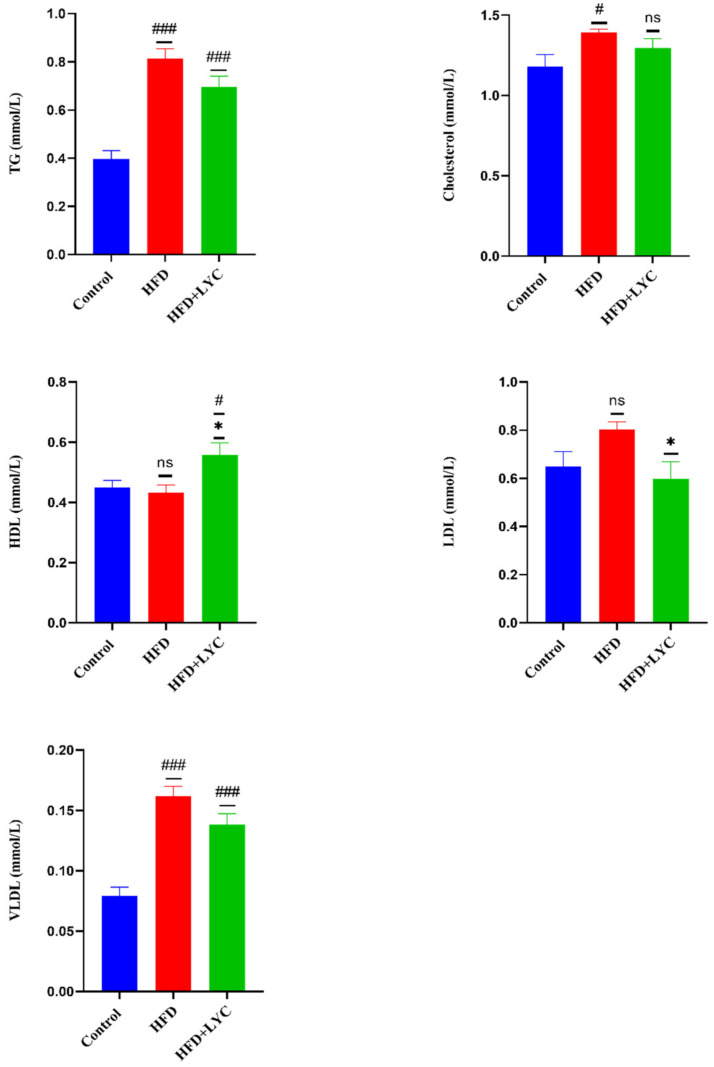
Serum levels of lipid profiles: TG, cholesterol, HDL, LDL, and VLDL, following treatment with LYC in rats fed an HFD. Data are expressed as the mean ± SEM of two replica (*n* = 12 rats). Different markers correspond to statistically significant differences between groups. # *p* < 0.05 and ### *p* < 0.001 indicate significant and very highly significant compared to the control, respectively. * *p* < 0.05 indicates a significant difference compared to the HFD group. ns: nonsignificant.

**Figure 3 molecules-27-07736-f003:**
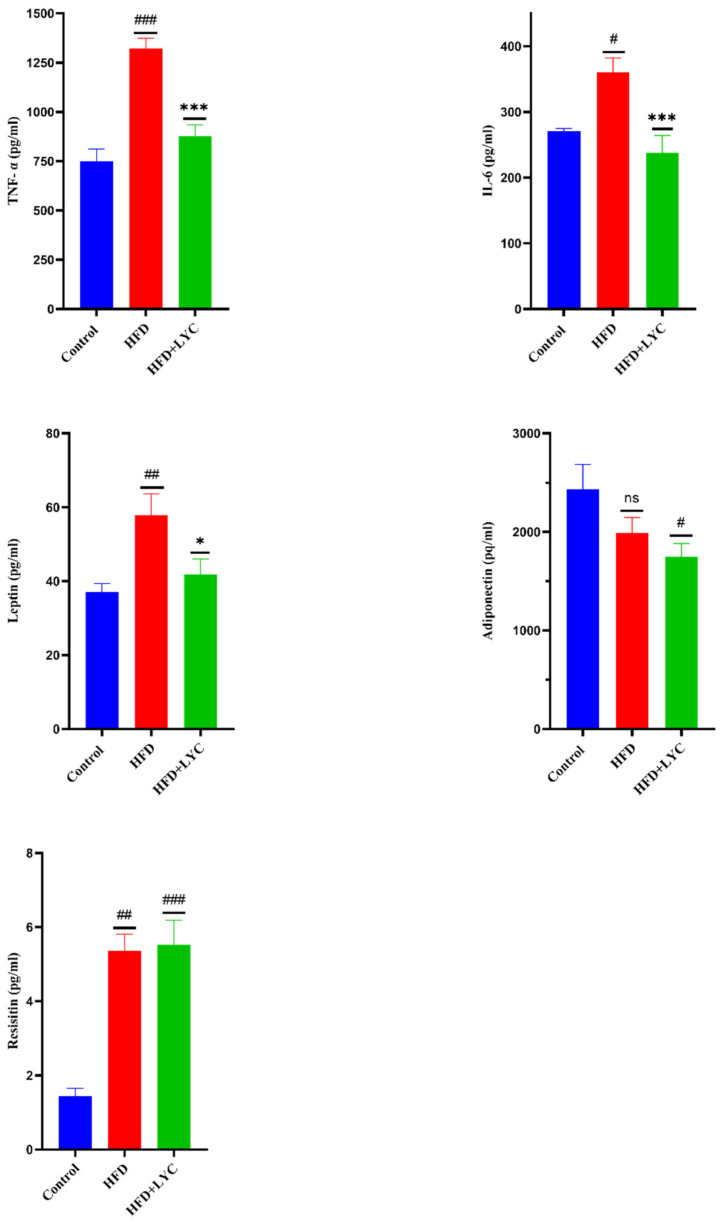
Hepatic levels of inflammatory markers TNF- α, IL-6, leptin, adiponectin, and resistin following treatment with LYC in rats fed an HFD. Data are expressed as the mean ± SEM of two replica (*n* = 12 rats). Different markers correspond to statistically significant difference between groups. # *p* < 0.05, ## *p* < 0.01, and ### *p* < 0.001 indicate significant, highly significant, and very highly significant differences compared with the control, respectively. ** p* < 0.05 indicates a significant difference, while *** *p* < 0.001 indicates a very highly significant difference compared to the HFD group. ns: nonsignificant.

**Figure 4 molecules-27-07736-f004:**
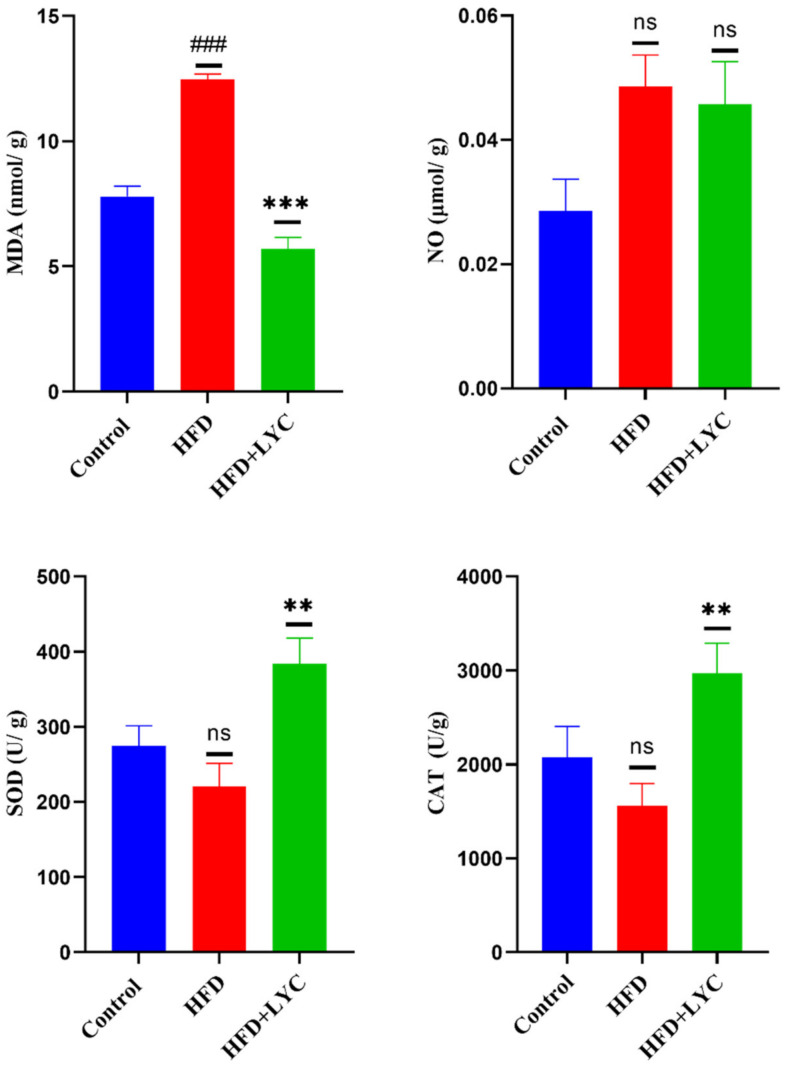
Hepatic levels of oxidative stress and antioxidant markers lipid peroxidation (MDA), nitric oxide (NO), superoxide dismutase (SOD), and catalase (CAT) following treatment with LYC in high-fat diet (HFD) rats. Data are expressed as the mean ± SEM of two replica (*n* = 12 rats). Different markers correspond to statistically significant differences between groups. ### *p* < 0.001: very highly significant compared with the control. ** *p* < 0.05 and *** *p* < 0.001 indicate a highly significant and a very highly significant difference compared to the HFD group. ns: nonsignificant.

**Figure 5 molecules-27-07736-f005:**
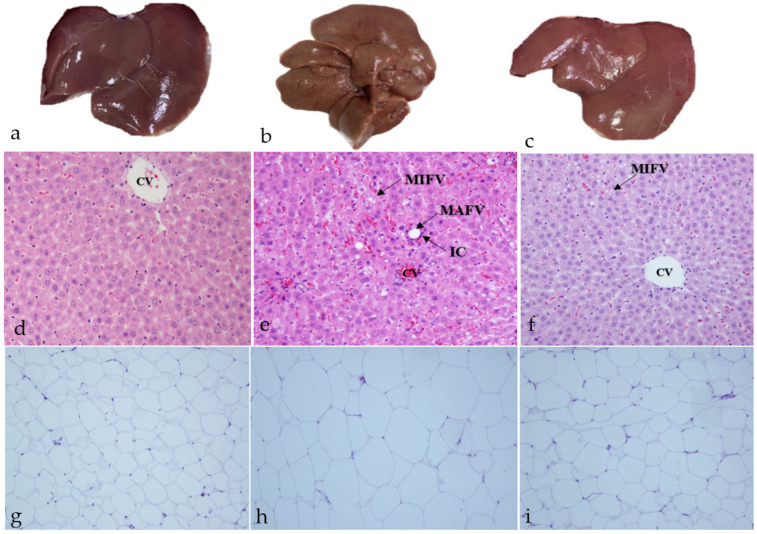
Liver gross anatomy and H&E of the control group (**a**,**d**), HFD group (**b**,**e**), and HFD + LYC group (**c**,**f**). All pictures were taken at 40× magnification. Histological changes in the WAT stained with H&E in (**g**) the control group, (**h**) the HFD group and (**i**) following treatment with LYC in HFD-induced obesity in rats, magnification was 20×.

**Table 1 molecules-27-07736-t001:** Food intake, body weight gain, liver weight, liver index, abdominal fat, and abdominal fat index following treatment with LYC (HFD + LYC rats).

Parameters	Control	HFD	HFD + LYC
Food intake (g/rat/day)	29.34 ± 0.35	23.61 ± 0.39 ^###^	23.26 ± 0.75 ^###^
Weight gain (%)	58.47 ± 4.62	104.94 ± 8.22 ^###^	109.29 ± 8.45 ^###^
Liver weight (g)	11.48 ± 0.30	11.58 ± 0.29 ^ns^	12.32 ± 0.40 ^NS^
Liver weight index (%)	3.12 ± 0.10	2.47 ± 0.06 ^###^	2.61 ± 0.13 ^##^
Abdominal fat (g)	4.70 ± 0.52	24.89 ± 2.51 ^###^	20.33 ± 1.35 ^###^
Abdominal fat index (%)	1.47 ± 0.20	5.23 ± 0.48 ^###^	4.02 ± 0.27 ^###,^ *

HFD: high-fat diet group; HFD + LYC: high-fat diet supplemented with lycopene. Data are expressed as the mean ± SEM (*n* = 12). Different markers correspond to statistically significant differences between groups at high significance (^##^
*p* < 0.01) and very high significance (^###^
*p* < 0.001) compared with the control; * *p* < 0.05: significant difference compared to the HFD group; NS: nonsignificant.

## Data Availability

Not applicable.

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
