# Peer review of "Lycopene Improves Metabolic Disorders and Liver Injury Induced by a Hight-Fat Diet in Obese Rats"

_molecules, 2022, doi:10.3390/molecules27227736_

Round 1
Reviewer 1 Report
The article is original, well structured; easy to read with the main emphasis on the antioxidant and anti-inflammatory potential of lycopene in obese rats via improving the metabolic disorders and liver injury induced via a high-fat diet. Overall the concept of the manuscript is significant however there are several places where unnecessary data is incorporated and the formatting is very poor. These don't detract from the meaning of the text but careful proofread could address these issues and improve the flow of the text. (For example: Line 209-210, Line 220-221, Line 229-231 is incomplete). The abstract needs to be redefined. The discussion section is not well elaborated and the language is not scientific. The conclusion is not at all consistent with the presented results. I would suggest the author using graph pad prism software to present the graphs as they will improve the efficacy of the manuscript. In my opinion, the manuscript can only be reconsidered after extensive language proofreading is done.
Author Response
Dear professor
Thank you for your valuable comments and your meticulous and thorough review.
Below is the response to your comments.
Point 1: The article is original, well structured; easy to read with the main emphasis on the antioxidant and anti-inflammatory potential of lycopene in obese rats via improving the metabolic disorders and liver injury induced via a high-fat diet. Overall the concept of the manuscript is significant however there are several places where unnecessary data is incorporated and the formatting is very poor.
Response1: Thank you for your valuable input, we have revised the data and improved the formatting.
Point 2: These don't detract from the meaning of the text but careful proofread could address these issues and improve the flow of the text. (For example: Line 209-210, Line 220-221, Line 229-231 is incomplete).
Response 2: The sentences in Lines 209-210, Line 220-221, Line 229-231 have been revised, rephrased, and completed.
Point 3: The abstract needs to be redefined.
Response 3: The abstract has been revised and improved.
Point 4: The discussion section is not well elaborated, and the language is not scientific.
Response 4: We have revised the discussion and it has also been edited by MPDI’s Language Editing Service.
Point 5: The conclusion is not at all consistent with the presented results.
Response 5: The conclusion has been entirely rewritten to appropriately address the findings.
Point 6: I would suggest the author using graph pad prism software to present the graphs as they will improve the efficacy of the manuscript.
Response 6: All the figures in the manuscript have been changed to ones generated by GraphPad Prism 9
Point 7: In my opinion, the manuscript can only be reconsidered after extensive language proofreading is done.
Response 7: I have sent the manuscript to MPDI’s Language Editing Service and received the certificate. I also kept all the changes made by the language editor tracked in the revised manuscripts.
Reviewer 2 Report
Comments
The present study considers Lycopene effect on controlling obesity and its adverse sequelae by assessing liver function parameters, lipid profile, inflammatory markers, oxidative stress biomarkers, and antioxidant enzymatic activities. Furthermore, a histopathological study was performed on rats fed with a Hight Fat Diet on the liver and white adipose tissue. The study is interesting; however; it needs to improve the following changes discussed below as well native person needs to revise the English
Line 1,2 : Lycopene Improves Metabolic Disorders and Liver Injury Induced by a Hight Fat Diet in Obese Rats “changed to” Lycopene Improves Metabolic Disorders and Liver Injury Induced by a High Fat Diet in Obese Rats
Line 13: Epidemiological studies have shown that consumption of a high fat diet (HFD) “here” use dash after high
Line 20: Antioxidants, lipid profile, liver function biomarkers and inflammatory markers “used” comma biomarkers
Line 28: High-fat diet “use” dash before fat
Line 34: Diabetes, hypertension, certain cancers “use” and after hypertension
Line 40: Dietary patterns, sedentary lifestyle, socioeconomic status, genetics “used” comma after genetics
Line 42-43: HFD induces calorie over consumption leading to weight gain and fat accumulation “here” use HFD induces calories
Line 48-55: Monitoring of markers of liver function, such as aspartate aminotransferase (AST), alanine aminotransferase (ALT), alkaline phosphatase (ALP), albumin, markers of lipid metabolism such as triglycerides (TG), cholesterol, high-density lipoprotein (HDL), low-density lipoprotein (LDL), very low-density lipoprotein (VLDL) and abdominal ultrasound are the most commonly used methods for assessing NAFLD risk “changed to” Monitoring of markers of liver function, such as aspartate aminotransferase (AST), alanine aminotransferase (ALT), alkaline phosphatase (ALP), albumin, markers of lipid metabolism such as triglycerides (TG), cholesterol, high-density lipoprotein (HDL), low-density lipoprotein (LDL), very low-density lipoprotein (VLDL) and abdominal ultrasound are the most commonly used methods for assessing various diseases including NAFLD risk “and” put references after various diseases such as 10.1111/jpn.13704; doi:https://doi.org/10.12681/jhvms.28497; and https://doi.org/10.3390/vaccines10010097
Line 59,60: These include leptin, adiponectin, tumor necrosis factor-alpha (TNF-α), interleukin 6 (IL-6) “ here” use comma after (IL-6)
Line 64: NAFLD clinical models evaluated “changed to” NAFLD clinical models were evaluated
Line 65: Superoxide dismutase (SOD) and catalase (CAT) in patients with NAFLD “add” comma after Superoxide dismutase (SOD)
Line 70: Obesity, cardiovascular disease and respiratory disease “add” comma after cardiovascular disease
Line 79: AST, ALT, ALP, albumin “add” comma after albumin
Line 81-84: Rephrase the sentence. Moreover, the modulating effect of LYC on reducing TNF-α, IL-6, leptin, adiponectin, resistin, oxidative stress markers such as MDA, NO and elevated the levels of antioxidant enzymes SOD, CAT in the livers of obese mice was reported by
Line 83- 84: Obese mice was reported by “here” add the references in the correct standard method of citation following journal format
Line 94-96: Rephrase the sentence. In contrast, HFD+LYC rats presented a significant decline (p < 0.05) in an abdominal fat index and no significant change in the aforementioned parameters relative to rats fed on HFD. Also, no significant difference was observed in liver weight between groups
Line 111-112: No significant change in albumin in HFD + LYC rats compared to the control group and HFD group “changed to” No significant change in albumin in HFD + LYC rats compared to the control and HFD groups
Line 138: Levels of hepatic TNF-α, IL-6, leptin and resistin exhibited notable increases “here” add comma after leptin
Line 139-140: While adiponectin was noticed no significant difference “changed to “ while adiponectin was noticed to have no significant difference
Line 155: No significant change in NO, SOD and CAT levels “here” add comma after SOD
Line 306-307: Rephrase the sentence
Line 335: Hence every 100 g of standard diet produced 387 kcal (271 + 80+ 36), 70% of calories derived from carbohydrates “changed to” Hence every 100 g of standard diet produced 387 kcal (271 + 80+ 36), 70% of calories are derived from carbohydrates,
Line 317, 410: In Material methods part, how many replicates/ samples were used for each different test? Write down in the Material and Methods part clearly
Line: 413: Oxidative stress, inflammation, and metabolic disorders “here” after inflammation and metabolic disorders add the comma
Line: 411-417: What about the future research that may be conducted following your findings?
Line 433-677: Carefully check the references, including the Journals name in the references part as well in the whole text following the journal standard references format
Author Response
Dear professor
Thank you for your valuable comments and for your meticulous and thorough review.
Below is the response to your comments.
Point 1: The present study considers Lycopene effect on controlling obesity and its adverse sequelae by assessing liver function parameters, lipid profile, inflammatory markers, oxidative stress biomarkers, and antioxidant enzymatic activities. Furthermore, a histopathological study was performed on rats fed with a Hight Fat Diet on the liver and white adipose tissue. The study is interesting; however; it needs to improve the following changes discussed below as well native person needs to revise the English
Response1: Thank you for your valuable comments, I have sent the manuscript to MPDI’s Language Editing Service and received the certificate. I also kept all the changes made by the language editor tracked in the revised manuscripts.
Point 2: Line 1,2: Lycopene Improves Metabolic Disorders and Liver Injury Induced by a Hight Fat Diet in Obese Rats “changed to” Lycopene Improves Metabolic Disorders and Liver Injury Induced by a High Fat Diet in Obese Rats
Point 3: Line 13: Epidemiological studies have shown that consumption of a high fat diet (HFD) “here” use dash after high
Point 4: Line 20: Antioxidants, lipid profile, liver function biomarkers and inflammatory markers “used” comma biomarkers
Point 5: Line 28: High-fat diet “use” dash before fat
Point 6: Line 34: Diabetes, hypertension, certain cancers “use” and after hypertension
Point 7: Line 40: Dietary patterns, sedentary lifestyle, socioeconomic status, genetics “used” comma after genetics
Point 8: Line 42-43: HFD induces calorie over consumption leading to weight gain and fat accumulation “here” use HFD induces calories
Response 2-8: All suggested changes have been addressed and corrected.
Point 9: Line 48-55: Monitoring of markers of liver function, such as aspartate aminotransferase (AST), alanine aminotransferase (ALT), alkaline phosphatase (ALP), albumin, markers of lipid metabolism such as triglycerides (TG), cholesterol, high-density lipoprotein (HDL), low-density lipoprotein (LDL), very low-density lipoprotein (VLDL) and abdominal ultrasound are the most commonly used methods for assessing NAFLD risk “changed to” Monitoring of markers of liver function, such as aspartate aminotransferase (AST), alanine aminotransferase (ALT), alkaline phosphatase (ALP), albumin, markers of lipid metabolism such as triglycerides (TG), cholesterol, high-density lipoprotein (HDL), low-density lipoprotein (LDL), very low-density lipoprotein (VLDL) and abdominal ultrasound are the most commonly used methods for assessing various diseases including NAFLD risk “and” put references after various diseases such as 10.1111/jpn.13704; doi:https://doi.org/10.12681/jhvms.28497; and https://doi.org/10.3390/vaccines10010097
Response 9: The sentence has been rephrased as suggested, and the citation has been added.
Point 10: Line 59,60: These include leptin, adiponectin, tumor necrosis factor-alpha (TNF-α), interleukin 6 (IL-6) “here” use comma after (IL-6)
Point 11: Line 64: NAFLD clinical models evaluated “changed to” NAFLD clinical models were evaluated
Point 12: Line 65: Superoxide dismutase (SOD) and catalase (CAT) in patients with NAFLD “add” comma after Superoxide dismutase (SOD)
Point 13: Line 70: Obesity, cardiovascular disease and respiratory disease “add” comma after cardiovascular disease
Point 14: Line 79: AST, ALT, ALP, albumin “add” comma after albumin
Point 15: Line 81-84: Rephrase the sentence. Moreover, the modulating effect of LYC on reducing TNF-α, IL-6, leptin, adiponectin, resistin, oxidative stress markers such as MDA, NO and elevated the levels of antioxidant enzymes SOD, CAT in the livers of obese mice was reported by
Point 16: Line 83- 84: Obese mice was reported by “here” add the references in the correct standard method of citation following journal format
Point 17: Line 94-96: Rephrase the sentence. In contrast, HFD+LYC rats presented a significant decline (p < 0.05) in an abdominal fat index and no significant change in the aforementioned parameters relative to rats fed on HFD. Also, no significant difference was observed in liver weight between groups
Point 18: Line 111-112: No significant change in albumin in HFD + LYC rats compared to the control group and HFD group “changed to” No significant change in albumin in HFD + LYC rats compared to the control and HFD groups
Point 19: Line 138: Levels of hepatic TNF-α, IL-6, leptin and resistin exhibited notable increases “here” add comma after leptin
Point 20: Line 139-140: While adiponectin was noticed no significant difference “changed to “ while adiponectin was noticed to have no significant difference
Point 21: Line 155: No significant change in NO, SOD and CAT levels “here” add comma after SOD
Point 22: Line 306-307: Rephrase the sentence
Point 23: Line 335: Hence every 100 g of standard diet produced 387 kcal (271 + 80+ 36), 70% of calories derived from carbohydrates “changed to” Hence every 100 g of standard diet produced 387 kcal (271 + 80+ 36), 70% of calories are derived from carbohydrates,
Response 10-23: All suggested changes have been addressed and corrected.
Point 24: Line 317, 410: In Material methods part, how many replicates/ samples were used for each different test? Write down in the Material and Methods part clearly
Response 24: The number of replicates has been added along with the number of animals tested is in the figure legends. The material and method part has been revised and improved.
Point 25: Line: 413: Oxidative stress, inflammation, and metabolic disorders “here” after inflammation and metabolic disorders add the comma
Response 25: comma has been added
Point 26: Line: 411-417: What about the future research that may be conducted following your findings?
Response 26: We have added the future prospective of our research at the end of the conclusion section.
Point 27: Line 433-677: Carefully check the references, including the Journals name in the references part as well in the whole text following the journal standard references format
Response 27: All the in-text citation and all references in the bibliography have been double checked.
Round 2
Reviewer 1 Report
This research has addressed the medicinal benefits of lycopene in the management of metabolic disorders. The authors have incorporated all the required suggestions. The title has also been improved and the abstract is in accordance with the title. Statistical significance is also clearly mentioned in the figure captions. The discussion is very well elaborated. The conclusion has also been reframed. Please remove the hyphen in the metabolism word in line 51. Altogether, I would suggest the erudite editor kindly accept this manuscript in its current form.
Reviewer 2 Report
Comments:
Line 49-53: Add more recent references as recommended
Line: 577-582: reference is repeated, removed it
Line: 606-608: Check the references carefully including Journals name used in the reference part following journal format